# *Aggregatibacter actinomycetemcomitans* Induces Autophagy in Human Junctional Epithelium Keratinocytes

**DOI:** 10.3390/cells9051221

**Published:** 2020-05-14

**Authors:** Emiliano Vicencio, Esteban M. Cordero, Bastián I. Cortés, Sebastián Palominos, Pedro Parra, Tania Mella, Constanza Henrríquez, Nelda Salazar, Gustavo Monasterio, Emilio A. Cafferata, Paola Murgas, Rolando Vernal, Cristian Cortez

**Affiliations:** 1Center for Genomics and Bioinformatics, Faculty of Science, Universidad Mayor, Camino la Pirámide 5750, Huechuraba 8580745, Chile; ev.fibonacci@gmail.com (E.V.); ecorv@hotmail.com (E.M.C.); sebastianpalominos.l@gmail.com (S.P.); pedro.parrar@mayor.cl (P.P.); tania.mella1@gmail.com (T.M.); constanza.hsoto@hotmail.com (C.H.); sofia.salazar.araya@gmail.com (N.S.); 2Center for Integrative Biology, Faculty of Science, Universidad Mayor, Camino la Pirámide 5750, Huechuraba 8580745, Chile; cortes.bastian@gmail.com (B.I.C.); paola.murgas@umayor.cl (P.M.); 3Parasitology Section, Instituto de Salud Pública de Chile, Avenida Marathon 1000, Ñuñoa 7780050, Chile; 4School of Dentistry, Faculty of Science, Universidad Mayor, Avenida Libertador Bernardo O’higgins 2013, Huechuraba 8580745, Chile; 5School of Medical Technology, Faculty of Science, Universidad Mayor, Camino la Pirámide 5750, Huechuraba 8580745, Chile; 6Periodontal Biology Laboratory, Faculty of Dentistry, Universidad de Chile, Sergio Livingstone Pohlhammer 943, Independencia 8380492, Chile; gmonasterio@ug.uchile.cl (G.M.); ecafferata@ucientifica.edu.pe (E.A.C.); 7Department of Periodontology, School of Dentistry, Universidad Científica del Sur, Av. Paseo de la República 5544, Lima 15074, Peru

**Keywords:** junctional epithelium keratinocytes, *A. actinomycetemcomitans*, LPS, autophagy, LC3, p62, cell viability

## Abstract

The adverse environmental conditions found in the periodontium during periodontitis pathogenesis stimulate local autophagy responses, mainly due to a continuous inflammatory response against the dysbiotic subgingival microbiome. The junctional epithelium represents the main site of the initial interaction between the host and the dysbiotic biofilm. Here, we investigated the role of autophagy in junctional epithelium keratinocytes (JEKs) in response to *Aggregatibacter actinomycetemcomitans* or its purified lipopolysaccharides (LPS). Immunofluorescence confocal analysis revealed an extensive nuclear translocation of transcription factor EB (TFEB) and consequently, an increase in autophagy markers and LC3-turnover assessed by immunoblotting and qRT-PCR. Correspondingly, challenged JEKs showed a punctuate cytosolic profile of LC3 protein contrasting with the diffuse distribution observed in untreated controls. Three-dimensional reconstructions of confocal images displayed a close association between intracellular bacteria and LC3-positive vesicles. Similarly, a close association between autophagic vesicles and the protein p62 was observed in challenged JEKs, indicating that p62 is the main adapter protein recruited during *A. actinomycetemcomitans* infection. Finally, the pharmacological inhibition of autophagy significantly increased the number of bacteria-infected cells as well as their death, similar to treatment with LPS. Our results indicate that *A. actinomycetemcomitans* infection induces autophagy in JEKs, and this homeostatic process has a cytoprotective effect on the host cells during the early stages of infection.

## 1. Introduction

Macroautophagy (hereafter referred to as autophagy) is a cellular homeostatic process that sequesters and delivers intracellular components to a lysosomal degradation/recycling pathway [1]. This cellular process has been investigated in numerous pathologies with heterogeneous etiologies, including inflammatory diseases of the oral cavity such as periodontitis [1,2,3]. Autophagy is an endogenous defense mechanism allowing cell survival during harsh conditions, such as starvation, misfolded proteins accumulation, hypoxia, endoplasmic reticulum (ER) stress, reactive oxygen species (ROS) accumulation, and infection [3,4]. The autophagy machinery comprises several autophagy-related genes (Atg) products and adaptor proteins, with 15 highly conserved core Atg proteins in mammals [1]. Autophagy begins when an isolation membrane sequesters and encloses cytosolic targets into a double-membrane vesicle, which subsequently elongates to form the autophagosome. Finally, the autophagosome fuses with lysosomes to form the autolysosome, leading to the degradation of its cargo [5,6]. The regulation of autophagosomes biogenesis is mainly attained by transcription factors, where transcription factor EB (TFEB) is the master regulator [7]. Autophagy is also essential in (i) the recognition, degradation, and selective elimination of bacterial pathogens (a phenomenon termed xenophagy), including *Streptococcus pyogenes*, *Mycobacterium tuberculosis*, *Salmonella* sp., and *Listeria monocytogenes*; as well as, in (ii) protection of the host cell from exotoxins and endotoxins released by pathogens, such as lipopolysaccharides (LPS) [4,8].

Periodontitis is a chronic inflammatory disease in which the dysbiotic subgingival microbiome and the concomitant host immune response plays a pivotal role in the aetiopathogenesis of the disease [9,10]. Periodontal lesions exhibit inflammation, connective attachment breakdown, and alveolar bone resorption, which lead to the migration of the junctional epithelium, formation of the periodontal pocket, and eventually, tooth loss [11]. As a component of the dysbiotic biofilm, *Aggregatibacter actinomycetemcomitans* has been closely related to the onset, progression, and severity of periodontitis, mainly due to its displayed virulence capabilities in damaged sites [12,13]. *A. actinomycetemcomitans* is a capnophilic Gram-negative rod, that has been classified into seven different serotypes (a–g) based on the structural and immunogenic differences of their LPS [14]. Serotype b has been found to be the most pathogenic and prevalent in periodontitis patients [15]. During the pathogenesis of periodontitis, *A. actinomycetemcomitans* use different virulence factors to colonize the gingival sulcus, infiltrate the junctional epithelium, invade and disseminate into the subjacent connective tissues, trigger host immune responses, and induce alveolar bone resorption [16,17,18]. To successfully establish periodontitis, *A. actinomycetemcomitans* must first break the front line of the host’s defense at the periodontium, where the junctional epithelium is critical in the initial bacteria/host interactions. Indeed, apart from acting as a mechanical barrier that shields deep periodontal tissues, the junctional epithelium is an important producer of antimicrobial agents and pro-inflammatory mediators [19]. In the context of periodontitis, autophagy is stimulated by adverse environmental conditions generated by the ROS accumulation in the periodontium due to a continuous inflammatory response against the dysbiotic biofilm [20,21]. These ROS-elevated levels trigger the intracellular signaling that stimulates either survival or death of infected cells through autophagy induction [22,23]. According to this, a positive correlation between high levels of ROS and increased expression of autophagy genes in peripheral blood mononuclear cells from periodontitis patients was detected [24]. The same authors reported the increased expression of autophagy proteins in human gingival fibroblasts treated with LPS of *Porphyromonas gingivalis* [24]. Furthermore, the treatment of these cells with antioxidants reduced the ROS concentration and consequently diminished autophagy, suggesting an autophagy-mediated modulation of periodontal inflammation.

Despite the previously reported involvement of autophagy in periodontitis, the role of this homeostatic process in the early junctional epithelium response during the infection with *A. actinomycetemcomitans* has not been clarified yet. This study aimed to analyze whether an infection with *A. actinomycetemcomitans* induces autophagy in the host cell and its role during the onset of periodontitis. Herein, we have developed an in vitro infection model using *A. actinomycetemcomitans* serotype b, its purified LPS, and junctional epithelium keratinocytes (JEKs) to recreate the initial stage of the pathogenesis of periodontitis.

## 2. Materials and Methods

### 2.1. Cell Line and Culture Conditions

Immortalized human gingival keratinocytes from the junctional epithelium (OKF6/TERT-2 cell line was obtained from Dr. Denisse Bravo) were maintained at 37 °C under a humidified atmosphere containing 5% CO_2_ in keratinocyte serum-free medium (#37010-022 Gibco, Carlsbad, CA, USA), supplemented with bovine pituitary extract (#13028-014, Gibco, Carlsbad, CA, USA), epidermal growth factor (#10450-013, Gibco), calcium chloride solution 0.3 M (#102382, Merck, Darmstadt, Germany), and 100 U/mL penicillin 100 µg/mL streptomycin (#15140-122, Gibco).

### 2.2. Bacterial Strain and LPS Purification

*A. actinomycetemcomitans* serotype b (ATCC^®^ 43718^™^) was used in this study. Bacteria were cultured in a capnophilic environment (8% O_2_ and 5%–12% CO_2_) at 37 °C in Brain Heart Infusion (BHI) medium (#CM1135B, Oxoid, Hampshire, UK) supplemented with 10% horse serum, as previously described [25,26]. To ensure viable bacteria expressing their full antigenic potential, all experiments were performed with bacteria harvested at the exponential growth phase. LPS from *A. actinomycetemcomitans* serotype b was purified by the Tri-reagent method and analyzed by 14% SDS-PAGE stained with periodic acid-silver, as previously described [27,28] (Appendix A).

### 2.3. In Vitro Infection Model

Cell infection assays were performed in 24-well plates (#174899, Thermo Fisher Scientific, Waltham, MA, USA) containing 13 mm diameter round glass coverslips at the bottom of each well. Coverslips were coated with 1.5 × 10^5^ adherent OKF6/TERT-2 (70%–80% confluence) in supplemented keratinocyte serum-free medium. Individual wells were washed with sterile PBS to remove the antibiotics and then incubated with bacteria (MOI = 200) diluted in antibiotic-free keratinocyte medium. Plates were centrifuged at 1200 rpm for 5 min and then incubated for 3 h at 37 °C. At the end of the incubation period, cells were washed thrice with sterile PBS, and the slides were processed for downstream assays. The same above approach was employed for LPS stimulation.

### 2.4. Total RNA Extraction and Reverse Transcriptase Polymerase Chain Reaction (RT-PCR)

Total RNA from untreated controls and experimental conditions was extracted using TRIzol™ Reagent (#15596026, Invitrogen, Carlsbad, CA, USA) according to the manufacturer’s instructions. RNA yield and integrity were determined by spectroscopy and gel electrophoresis, respectively. Target-transcript levels were analyzed in StepOnePlus™ Real-Time PCR System (#4376600, Applied Biosystems^®^, Foster City, CA, USA) using KAPA SYBR FAST qPCR Master reagent (Kappa Biosystems^®^, Sigma-Aldrich Corporation, St. Louis, MO, USA). The sequences of primers used in this study were as follows: TFEB forward (F) 5′-GGTGCAGTCCTACCTGGAGA-3′ and reverse (R) 5′-GTGGGCAGCAAACTTGTTCC-3′; ATG5, (F) 5′-TTTGCATCACCTCTGCTTTC-3′ and (R) 5′-TAGGCCAAAGGTTTCAGCTT-3′; LC3B, (F) 5′-GAGAAGCAGCTTCCTGTTCTGG-3′ and (R) 5′-GTGTCCGTTCACCAACAGGAAG-3′; p62, (F) 5′-TGCCCAGACTACGACTTGTG-3′ and (R) 5′-AGTGTCCGTGTTTCACCTTCC-3′; 18S rRNA, (F) 5′-GTAACCCGTTGAACCCCATT-3′ and (R) 5′-CCATCCAATCGGTAGTAGCG-3′. Each sample was run in triplicate, and the relative amount of target transcripts was quantified by the 2^−ΔΔCt^ method after normalization to the 18S mRNA levels.

### 2.5. Immunoblot Assays

Untreated control cells, LPS-stimulated or bacteria-infected JEKs were homogenized in TNE buffer (0.1 M Tris-HCl, 0.25 M NaCl, and 0.05 M EDTA) supplemented with protease inhibitor cocktail (#P8340, Sigma-Aldrich, St. Louis, MO, USA) and the protein concentration was determined by BCA assay (#23225, Pierce™ BCA Protein Assay Kit, Thermo Scientific, Waltham, MA, USA), according to the manufacturer’s instructions. Cell lysates were resolved by SDS-PAGE using Mini-PROTEAN^®^ system (Bio-Rad, Hercules, CA, USA), transferred to nitrocellulose membranes, and incubated with antibodies specific to LC3B (mAb D11 #3868, Cell Signaling Technology, Danvers, MA, USA), Atg5-Atg12 (#2630, Cell Signaling Technology), p62 (#ab56416, Abcam, Cambridge, UK), and β-actin (#4970 mAb 13E5, Cell Signaling Technology). Membranes were developed using Immobilon Western Substrate (WBKL S0-500, Millipore Corporation, Burlington, MA, USA), and the chemiluminescent signal was acquired using ChemiDoc™ XRS+ System with Image Lab™ software (#1708265 Bio-Rad, Hercules, CA, USA). The intensity of bands was measured by densitometry using ImageJ software (National Institute of Mental Health, 2010). The graph shows the average of the densitometric analysis (actin normalization) from three independent experiments.

### 2.6. Indirect Immunofluorescence Confocal Assays

Assays were performed, as previously detailed [29]. Briefly, coverslips with 1.5 × 10^5^ adherent OKF6/TERT-2 cells were incubated with *A. actinomycetemcomitans* or LPS for 3 h at 37 °C. Cells were then washed with PBS to remove unbound bacteria or LPS, fixed with 4% p-formaldehyde (PFA) in PBS for 30 min, and quenched with 50 mM NH_4_Cl in PBS for 30 min. Coverslips were washed with PBS and incubated for 2 h at room temperature with rabbit anti-human LC3B (mAb D11), diluted 1:100 (v/v) in a PBS solution containing 0.15% gelatin, 0.1% sodium azide, and 1% saponin (PGN-Saponin). Coverslips were washed with PBS and incubated for 1 h with Alexa Fluor 488-conjugated anti-rabbit IgG (#A11008, Invitrogen), diluted 1:300 in PGN-Saponin. After PBS washes, the coverslips were incubated with anti-LPS antibody (#PA1-73178, Invitrogen), diluted 1:200 in PGN-Saponin, washes again and incubated for 1 h with Alexa Fluor 555-conjugated anti-goat IgG (#A21422, Invitrogen), diluted 1:500 in PGN-Saponin containing 10 µg/mL DAPI (4′,6′-diamidino-2-phenylindole dihydrochloride, #D1306, Thermo Fisher Scientific). The same procedure was employed for the preparation of samples stained with the following antibodies: anti-TFEB (#ab122910, Abcam, Cambridge, UK) 1:100, anti-p62 (#ab56416) 1:100, anti-LAMP2 (H4B4, from Developmental Studies of the Hybridoma Bank, University of Iowa) 1:12. Alternatively, cell-coated slides were incubated with antibodies directed against β-tubulin (1:500) conjugated to Alexa Fluor-555 or phalloidin-Alexa Fluor-568 (1:1000) (#8878, Cell Signaling Technology) to visualize the cellular contours. Finally, samples were mounted using ProLong^®^ Gold antifade reagent mountant (#P36930, Life Technologies) and analyzed in a Leica TCS SP8 confocal laser scanning microscope (Leica Microsystem, Wetzlar, Germany) using an oil immersion Plan-Apochromat 63× objective (numerical aperture 1.4). The series of images obtained from confocal z-stacks were processed and analyzed using Leica LAS AF (Leica Microsystem 2012, Wetzlar, Germany) and Imaris software (Bitplane, Belfast, UK).

### 2.7. Immunofluorescence Colocalization Analysis

Confocal images of untreated control cells or LPS treated and infected keratinocytes, immunostained with anti-TFEB (#ab122910), and DAPI were processed for colocalization analysis using the *coloc* tool of Imaris software, as previously reported [29]. Briefly, the software allowed the construction of additional fluorescent channels corresponding to colocalized voxels adjusted by fluorescent thresholds. TFEB fluorescent channel was colocalized with the DAPI channel, which was depicted as a new image of colocalized voxels. The software quantified the number of colocalized voxels between the channels (DAPI/TFEB) by applying the same thresholds for all experimental conditions.

### 2.8. Algorithm-Based Autophagosomes Detection

Detection was performed as previously reported [29], with some modifications. The series of z-stack images acquired at confocal microscopy were processed by Imaris software for three-dimensional (3D) reconstructions and detection of autophagosomes in x-y-z coordinates. Autophagosomes detection was performed as follows. Confocal images of OKF6/TERT-2 cells were immunostained with anti-LC3B (autophagosome), anti-LPS (bacteria), and DAPI (nucleus), and then processed by Imaris software, which allowed the construction of isospots from the fluorescence signals [30,31]. Isospots were constructed based on two classes of LC3B signal detection, colocalizing or not with intracellular bacteria; thus, generating two types of isospots (green and purple spheres) with different sizes.

### 2.9. Lysosomes Alkalinization and Inhibition of the Downstream Autophagic Activity

Lysosomal alkalinization was performed as reported elsewhere [32]. Briefly, OKF6/TERT-2 cells were incubated for 1 h with 200 nM bafilomycin A1 (Alexis Biochemicals, CA, USA), 10 µM chloroquine diphosphate salt (#C6628, Sigma-Aldrich), or 20 mM ammonium chloride (NH_4_Cl) (#254134, Sigma-Aldrich) to inhibit the downstream autophagic activity [5]. After washing out the autophagy inhibitors, the cells were incubated with 1 µM of acridine orange (#14338, Cayman chemical, Ann Arbor, MI, USA) for 20 min at 37 °C to evaluate the alkalinization of lysosomes (Appendix A). The acridine orange is a permeable fluorescent cationic dichromatic probe that detects the pH-dependent changes in mammalian cells. This assay allows the detection of emission spectra of 525 nm (green) and 650 nm (red). The emission spectra correspond to the basification (green) and acidification (lysosomes in red) processes of the cell structures after the treatment with downstream autophagy inhibitors [33]. The samples were analyzed in the CytoFLEX^TM^ V3-B3-R3 flow cytometer (#B53007, Beckman-Coulter Life Sciences, Brea, CA, USA) in PerCP detector (Ex-Max 488 nm/Em-Max 650–690 nm). The acquired data were analyzed using FlowJo v10.0.8 software (Tree Star Inc., Ashland, OR, USA).

### 2.10. Cell Viability Analysis

OKF6/TERT-2 cells were incubated with 3 mM 3-Methyladenine (3-MA) (#ab12084, Abcam), an upstream autophagy inhibitor, for 3 h and then stimulated with *A. actinomycetemcomitans* (MOI = 200) or 1 µg/mL of its LPS for 3 h at 37 °C. Cells were washed and incubated for 20 min with Zombie Aqua™ Fixable Viability Kit (#423101, Biolegend^®^, San Diego, CA, USA), an amine-reactive fluorescent dye that is non-permeant to live cells but permeant to cells with compromised membranes. Samples were washed, fixed in 4% PFA, and resuspended in PBS prior to analysis in CytoFLEX^TM^ V3-B3-R3 flow cytometer (Beckman-Coulter Life Sciences). Fluorescence intensity and cell counts were determined using the pacific blue detector, acquiring a total of 10^5^ events. Collected data were analyzed using FlowJo v10.0.8 software (Tree Star Inc., Ashland, OR, USA).

Cell viability was further analyzed by a second approach. Briefly, cells were treated and stimulated as aforementioned, followed by incubation for 15 min with the fluorescent DNA-binding dyes DAPI and 7-Aminoactinomycin D (7-AAD) (#A1310, Life Technologies), which are semipermeable and non-permeable to live cells, respectively. Following incubation, samples were washed, fixed with 4% PFA, mounted using Prolong Gold antifade (#P36930, Life Technologies,) and analyzed by confocal microscopy. The percentage of viable cells was calculated subtracting the number of dead cells (7-AAD positive) from the total number of cells (DAPI positive).

### 2.11. Statistical Analysis

The significance of differences observed between experimental conditions was determined by the unpaired *t*-test using GraphPad Prism 7.0 package (GraphPad software, Inc., CA, USA). The level of significance was set at *p* < 0.05.

## 3. Results

### 3.1. A. Actinomycetemcomitans Induces Autophagy in JEKs

*A. actinomycetemcomitans* has been strongly implicated in the development of rapidly progressing periodontal disease due to its ability to adhere, invade, and damage the junctional epithelium, adjacent to the tooth surface. [16,17,18]. To evaluate whether *A. actinomycetemcomitans* induces autophagy in this periodontal context, JEKs were incubated with *A. actinomycetemcomitans* (Figure 1) and then processed using confocal immunofluorescence to visualize TFEB. Upon autophagy stimulation, TFEB translocates from the cytoplasm to the nucleus, where it induces the transcription of genes involved in autophagosomes biogenesis [34].

JEKs challenged with *A. actinomycetemcomitans* showed an extensive nuclear translocation of TFEB (Figure 2A) as determined by the quantification of colocalized DAPI/TFEB voxels (Figure 2B). Consistently, increased levels of TFEB transcripts were detected in challenged cells (Figure 2C). Likewise, with the activation of TFEB, we observed an increase of Atg5 transcription (Figure 2D) and Atg5-Atg12 protein complex expression in infected JEKs (Figure 2E, and Appendix A). Microtubule-associated protein 1 light chain 3 (LC3B-I) is a soluble cytosolic protein that is converted into LC3B-II, on the autophagosome membrane, upon autophagy stimulation [5,35]. Although no differences in LC3B transcription levels were detected in OKF6/TERT-2 cells (Figure 2F), increased LC3B-II expression was observed after *A. actinomycetemcomitans*-challenge, suggesting a bacterial-induced LC3B turnover (Figure 2G, Figure 5B, and Appendix A).

Confocal images revealed enhanced expression of LC3B protein in infected keratinocytes, indicating a bacteria-induced autophagosome formation (Figure 3A, yellow arrowhead, and Figure 3B). Moreover, three-dimensional (3D) reconstructions of confocal sections [29,31] showed a close association between intracellular bacteria and LC3B-positives vesicles (inset of Figure 3A, Figure 3B–D, and Appendix A), suggesting that *A. actinomycetemcomitans* exploits autophagy route during its intracellular life cycle. Two populations of LC3B-positive vesicles were detected, vesicles of approximately 2.5 μm containing bacteria (Figure 3E,F, green spheres) and a population of about 1.2 μm without bacteria (Figure 3E, purple spheres). We speculate that the vesicles of 1.2 μm could be induced by the ability of *A. actinomycetemcomitans* to secrete many LPS-coated outer membrane vesicles (OMVs) during infection, enhancing its virulence and exacerbating the host’s inflammatory response [14,18]. Taken together, these results suggest that *A. actinomycetemcomitans* induces autophagy in JEKs.

### 3.2. Purified LPS from A. Actinomycetemcomitans Induces TFEB Nuclear Translocation and Biogenesis of LC3-Positive Vesicles in JEKs

To elucidate whether the LPS released from *A. actinomycetemcomitans* during the infection is able to induce autophagy in the host cell, JEKs were incubated with purified LPS and then analyzed by confocal microscopy. Extensive nuclear translocation of TFEB, independent of LPS concentration, was observed (Figure 4A,B). Quantification of colocalized DAPI/TFEB voxels in LPS kinetic-stimulation assays confirmed that LPS-induced TFEB activation was time-dependent (Figure 4C (0.5 µg/mL) and 4D (1 µg/mL)). Nevertheless, TFEB and LC3B transcriptional levels were increased only in cells stimulated with 1 µg/mL of purified LPS (Figure 4E and Figure 5A). Consistent with TFEB activation, *A. actinomycetemcomitans*’ LPS induced a strong turnover of LC3B protein (Figure 5B). Densitometric measurements showed a significant increase in LC3B-II expression at 3 h after LPS-challenge, an effect that was inhibited in keratinocytes pretreated with 3-MA, an inhibitor of autophagosomes biogenesis (Figure 5C,D). In the same context, JEKs were incubated for 3 h with 1 µg/mL of purified LPS and then subjected to confocal immunofluorescence analysis for detection and visualization of autophagosomes and endocytosed LPS. Confocal images showed a punctuate cytosolic profile of LC3B protein in LPS-challenged cells, in contrast to the diffuse distribution observed in untreated controls (Figure 5E). Similarly, the vesicular profile of LC3 (autophagosomes) was intimately associated with the cytosolic presence of LPS, as shown in Figure 5E (LC3B/LPS colocalization channel), suggesting that LC3B-vesicular profile (LC3-II) was induced by endocytosed-LPS. Overall, these results indicate that LPS from *A. actinomycetemcomitans* induces autophagy in JEKs.

### 3.3. A. Actinomycetemcomitans Induces Selective Autophagy Mediated by p62/SQSTM1

Ubiquitin is a crucial molecule in xenophagy that labels substrates that will undergo selective degradation. Autophagy adaptors, such as p62/SQSTM162 (p62), have ubiquitin-binding domains and LC3-interacting regions that recognize ubiquitinated substrates and deliver them for lysosomal degradation. P62 is an adapter protein that has been associated with anti-bacterial and selective autophagy induced by LPS [4,36]. Thus, we investigated the influence of bacteria or purified LPS on p62-recruitment in JEKs. Although infected cells did not show significant differences in p62-expression as compared with the non-infected control (Figure 6A,B and Appendix A), an intimate association between p62-labeling in LC3B-positive vesicles containing bacteria was observed (Figure 6C, white arrowheads). Approximately 62% of the LC3B-positive bacteria also exhibited p62-labeling (Figure 6C graph). Likewise, JEKs incubated with LPS (1 µg/mL) exhibited a significant increase in the transcription and expression of p62 (Figure 6D,E). Confocal microscopy images corroborated the colocalization between p62 and endocytosed-LPS (3 h after challenge) (Figure 6F). Taken together, these results indicate that p62 is the main adapter protein recruited during *A. actinomycetemcomitans* infection.

### 3.4. The Pharmacological Inhibition of Autophagy Increases the Infected Cell Number and the Cell-Death of A. Actinomycetemcomitans-Challenged JEKs

Autophagy is a sequential dynamic process that involves first the biogenesis and acidification of vesicles, followed by the degradation of vesicular contents after their fusion with lysosomes, which is the reason why this process can be pharmacologically inhibited in different stages [1,5]. To assess the role of JEKs-autophagy during *A. actinomycetemcomitans* infection, we inhibited autophagic flux in early and late stages. First, OKF6/TERT-2 cells were challenged for 3 h with *A. actinomycetemcomitans* and subsequently treated with 200 nM bafilomycin A1, 10 µM chloroquine, or 20 mM NH_4_Cl (Figure 7A). These compounds alkalinize the host’s lysosomes (Appendix A), inhibiting the maturation of autophagosomes to autolysosomes, a fundamental step in the degradation of autophagic cargoes [5]. Downstream-autophagy inhibition significantly increased the number of infected cells (Figure 7B, central column, and Figure 7C). In addition, many intracellular bacteria were observed closely colocalizing with the host-lysosomes in cells treated with autophagic inhibitors (Figure 7B, yellow arrowheads in the left column). Consistent with the increase of infected cells, treated-JEKs exhibited several bacteria colocalizing with intercellular actin protrusions (Figure 7D white arrowheads). These suggest that the arrest of autolysosomal degradation increased intracellular-bacterial accumulation and favored the transcellular spread. In the same context, autophagy can play a dual role in periodontitis, either promoting or blocking cell-death, depending on the host cell type [21].

We also evaluated the effect of upstream autophagy pharmacological inhibition on the viability of infected or LPS-treated keratinocytes. JEKs were pretreated with 3-MA (3 h) and then challenged with *A. actinomycetemcomitans* or their purified-LPS (3 h). First, we checked that the pretreatment with the drug does not affect the number of infected JEKs by the bacteria (Appendix A). 3-MA blocks the formation of autophagosomes by inhibiting the class III phosphatidylinositol 3-kinases (PI3K) [35]. Cell viability was assessed by two different methodologies: confocal microscopy (Figure 8A) and flow cytometry (Figure 8B), as specified in the methods section. Both approaches revealed a significant increase in JEKs death when upstream autophagy was suppressed prior to challenge with *A. actinomycetemcomitans* or its purified LPS (Figure 8A,B graph), suggesting that autophagy has a protective effect against cell-death during *A. actinomycetemcomitans* infection. Collectively, our results indicate that *A. actinomycetemcomitans* and its LPS induce autophagy in JEKs, a molecular process that has a protective effect on host cells in the early stage of the infection by this periodontal pathogen.

## 4. Discussion

The defense mechanism provided by the junctional epithelium is key for protection against the onset of periodontitis and the formation of the periodontal pocket. Indeed, structural changes are pivotal in the transformation of the junctional epithelium to a pocket epithelium and determine the formation of the periodontal pocket [37]. In this context, protective mechanisms displayed by JEKs, such as autophagy, are critical for periodontal protection against bacteria and their products. Autophagy can protect cells from apoptosis; however, excessive autophagy can destroy essential cellular components and lead to cell death [22]. In the present study, we demonstrated for the first time that *A. actinomycetemcomitans* infection induces autophagy in human JEKs, a cellular homeostatic process with a cytoprotective effect on this cell type, in the early stages of infection (Figure 9).

During periodontitis pathogenesis, the junctional epithelium represents the first natural barrier that biofilm-forming bacteria must undermine to reach deeper periodontal tissues. As an interface between the gingival sulcus and connective tissues, the junctional epithelium controls the constant microbiological challenge, protecting the tooth-supporting periodontium [19,38]. Our analysis revealed that the challenge with bacteria or its purified LPS induced autophagy in OKF6/TERT-2 cells during the first hours after stimulation. Several lines of evidence suggest that the activation of TFEB early during infection is evolutionarily conserved, positioning this molecule as a key component of host defense [39]. Immune signaling elicits inflammation by promoting cellular damage, and TFEB has a recognized anti-inflammatory protective activity against microbial stimuli or its secreted products, including LPS. In this context, TFEB suppresses inflammation in a manner dependent or independent of the Nuclear Factor-kappa B (NF-kB) modulation, a key signaling pathway for the maintenance of immune homeostasis of the epithelia. [39,40,41,42]. Phosphorylated TFEB is localized on the lysosomes in the cytosol; however, when it is dephosphorylated, it translocates to the nucleus and stimulates the expression of genes involved in lysosomal biogenesis and autophagy induction [34]. In this study, bacterial and LPS challenge induced an increase in transcription of TFEB mRNA and extensive nuclear translocation of TFEB protein to the nucleus of JEKs (Figure 2A,B and Figure 4), where it acts as the primary regulator of the autophagosomes biogenesis [34]. Consistently, stimulated cells also exhibited a significant increase in essential autophagic markers involved in the early stages of autophagosomes formation. Bacteria and its purified LPS induced a significant LC3-protein turnover, a widely used parameter for evaluation/confirmation of autophagic flux (Figure 2D–G and Figure 5A–D) [5,35]. Confocal microscopy analysis confirmed a vesicular profile of LC3 protein in challenged cells, in contrast to the diffuse pattern exhibited by untreated cells (Figure 3A and Figure 5E). 3D reconstructions of infected JEKs showed two populations of LC3-positive vesicles with different sizes, i.e., vesicles of 2.5 µm containing bacteria and smaller LPS-positive vesicles without bacteria inside. This suggests that both, intracellular bacteria and endocytosed-LPS released during infection activated the autophagic pathway. Nevertheless, because the molecular processes that induce dephosphorylation and activation of TFEB are quite complex, the mechanisms by which *A. actinomycetemcomitans* and its LPS dephosphorylate TFEB and induce its migration to the nucleus needs to be fully clarified. Xenophagy and LPS-induced autophagy are selective processes. Adaptor-mediated LC3 recruitment is a widely accepted model for explaining the selective recognition of ubiquitinated substrates by the autophagic machinery [4]. Our study confirmed the role of p62 as the main autophagy-adapter protein during the early stage of *A. actinomycetemcomitans* infection. The human p62 protein contains an N-terminal LIR (LC3-interacting region) motif and a C-terminal UBA (Ubiquitin-associated) domain [36]. In accordance with this, we observed that approximately 62% of the LC3-positive bacteria also exhibited p62-labeling, which suggests that p62 links the bacteria to autophagosomes via LC3 (Figure 6). Collectively, these data suggest that the selective autophagy stimulated in gingival keratinocytes was in response to the initial interaction with *A. actinomycetemcomitans* and their products.

Although the processes of adhesion/invasion, capture, and microbial degradation have been well characterized, the host mechanisms that recognize intracellular pathogens to induce xenophagy remain unclear [4,8]. Based on our results, we visualized a plausible scenario that was schematically represented in Figure 9 (Upper panel). *A. actinomycetemcomitans* adheres and invades epithelial cells through a process that involves actin polymerization and subsequent receptor-mediated endocytosis [16]. In parallel, LPS-coated OMVs are actively released by *A. actinomycetemcomitans* in vivo and can be internalized into cultured epithelial cells mainly via clathrin-dependent endocytosis or fusion with host cell membranes in a cholesterol-dependent manner [18]. According to our proposal, the bacteria, its purified LPS, or OMVs lead to the activation of NF-kB when recognized by the surface Toll-like receptors [17,43] or cytosolic NOD-like receptors [44,45]. Thus, the activated NF-kB pathway would trigger the dephosphorylation and activation of TFEB, which in turn induces its migration to the nucleus and the upregulation of autophagy-related genes. Once internalized, this bacterium has several virulence factors (such as hemolytic factors or phospholipase C) that allow it to transgress the endosomal membrane, and be released into the nutrient-rich cytosol, to begin its intracellular replication [16]. As a consequence, the damage to the endosomal membrane results in its ubiquitination and recruitment of p62, possibly through his C-terminal UBA domain. Once the vesicular cargo has been recognized, p62 interacts with the lipidized form of LC3 on the surface of the forming autophagosome (LC3-II) and directs *A. actinomycetemcomitans* and its LPS to lysosomal degradation through the autophagic pathway. Given the importance of these processes in antibacterial cell clearance, the mechanisms that trigger the ubiquitination of *A. actinomycetemcomitans* and its LPS, and the subsequent recruitment of p62, should be extensively addressed and clarified by future studies.

Evidence indicates that *A. actinomycetemcomitans* traverses the gingival epithelium using pro-apoptotic mechanisms triggered by the host’s enhanced pro-inflammatory response [17]. In the present study, the up and downstream pharmacological inhibition of autophagy (Figure 7 and Figure 8) significantly increased the infected-cell percentage and the cell death of *A. actinomycetemcomitans*-challenged or LPS-treated JEKs (Figure 9, lower panel). These findings confirm the pro-survival protective role of autophagy in periodontal tissues during the pathogenesis of periodontitis, which has been widely reported [23]. For instance, a recent study reported that macrophages infected with *A. actinomycetemcomitans* when treated with trans-cinnamic aldehyde significantly diminished the inflammatory response and induced bacterial-intracellular killing by autophagy activation [46]. Accordingly, the anti-inflammatory drug meresin inhibited the LPS-induced apoptotic cell-death of periodontal ligament cells via autophagy stimulation [47]. Exfoliative sulcular keratinocytes and human cultured keratinocyte cells (HaCaT) showed increased autophagy after *P. gingivalis*-LPS treatment, via increased-ROS levels [48]. A similar effect was observed in human periodontal ligament cells, in which increased autophagy was observed in response to hypoxia-induced apoptosis [22].

In agreement with other models of infection, our results indicate that JEKs autophagy is an important cell homeostatic mechanism of epithelial antibacterial defense that is activated by invasive bacteria and their products, in order to avoid the microbial dissemination and favor bacterial clearance [49,50,51]. In this context, LAMP-2-deficient mice, a key protein for the autolysosome’s formation, developed severe periodontitis in the early stages of life. These animals exhibited exacerbated accumulation of bacterial biofilm, gingival inflammation, alveolar bone resorption, loss of periodontal attachment, and apical migration of junctional epithelium, reinforcing the importance of bacterial-autophagic clearance of our results [52]. Contrary to our results with *A. actinomycetemcomitans*, *P. gingivalis*-induced autophagy has pro-bacterial effects. This important periodontal pathogen stimulates their entry into the autophagic pathway and, thus, avoids or delays their lysosomal degradation, increasing its colonization and penetration into host periodontal tissues, critical events for the spreading of the infection [53,54,55]. If the activation and differential modulation of the autophagic pathway by periodontal pathogens is important for the establishment and the severity of periodontitis, this would be an important topic that needs further clarification. In this regard, a recent report shows increased levels of autophagy-related proteins in gingival tissues of aggressive periodontitis patients, suggesting a potential distinctive role of autophagy among aggressive and chronic forms of the disease [56]. The authors also described increased autophagy activity in *A. actinomycetemcomitans*-infected THP-1-derived macrophages, generating anti-inflammatory consequences linked to ROS modulation [56]. Nevertheless, it is important to point out that these results were obtained with a cell type that requires external stimuli for its in vitro differentiation, which could generate some artifacts in the induction of the autophagy process. Given that the here presented research was carried out using a human periodontal keratinocyte cell line, our results can be interpreted closer to the periodontal physiological reality.

During infection, host cells respond to bacterial pathogens through the upregulation of degradative processes, including autophagy [57]. This enhancement was demonstrated here in *A. actinomycetemcomitans*-induced JEKs, suggesting a protective role of periodontium against infection, at least against this bacterial species. As the junctional epithelium consists of 1–3 cell layers in thickness at its apical termination [38], the antibacterial and pro-survival effect mediated by autophagy herein described, could provide valuable insight into the pathogenesis of periodontitis, periodontal health, and healing. In this context, the stimulation of autophagy could increase antibacterial resistance and tissue tolerance mechanisms, reducing the effects of the constant challenge of biofilm bacteria and their products on the junctional epithelium; thereby, preventing bacterial invasion, transcellular dissemination, and subsequent destructive immunoinflammatory response displayed during the onset and progression of periodontitis pathogenesis. In summary, our data highlight the critical relationships between periodontal infection and autophagy in gingival keratinocytes and reveal putative mechanisms implied in gingival protection and periodontitis prevention.

## 5. Conclusions

Autophagy is a key modulator of several molecular processes involved in inflammatory diseases, including periodontitis. These findings allow us to conclude that autophagy modulation and targeting within the epithelial gingival barriers could represent a novel therapeutic strategy for prevention of the early stages of periodontitis, possibly stopping its progression and promoting periodontal homeostasis.

## Figures and Tables

**Figure 1 cells-09-01221-f001:**
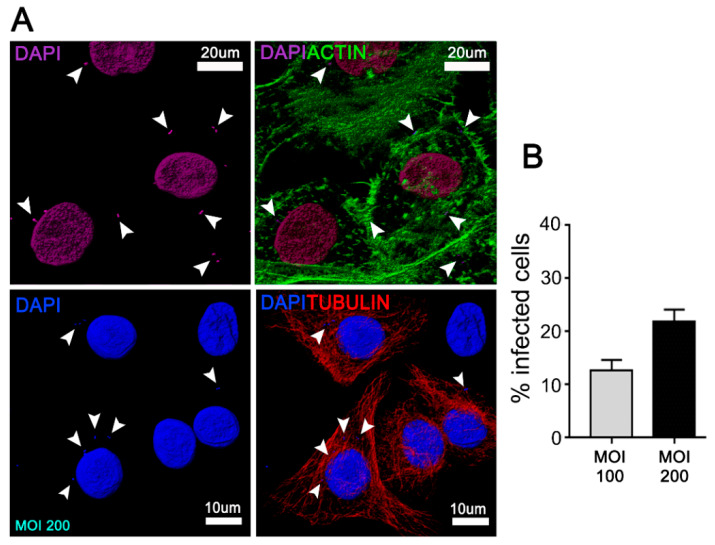
Junctional epithelium keratinocytes (JEKs)-invasion by *A**ggregatibacter actinomycetemcomitans* (*Aa*). OKF6/TERT-2 cells were incubated with *A.a* (serotype b) at MOI = 100 and 200 for 3 h at 37 °C. After the corresponding washes, JEKs were fixed and subjected to confocal microscopy analysis for the quantification of infected cells. (**A**) Immunofluorescence staining using actin (upper panel in green) and tubulin (bottom panel in red) to delineate cell contour, as well as DAPI (purple, upper panel and blue, bottom panel) for visualization of cell nuclei and bacteria. White arrowheads indicate intracellular bacteria. (**B**) The graph shows the average of 15 images acquired by confocal microscopy using a 63× objective in five independent experiments (≥1000 cells). Scale bar 10 and 20 µm.

**Figure 2 cells-09-01221-f002:**
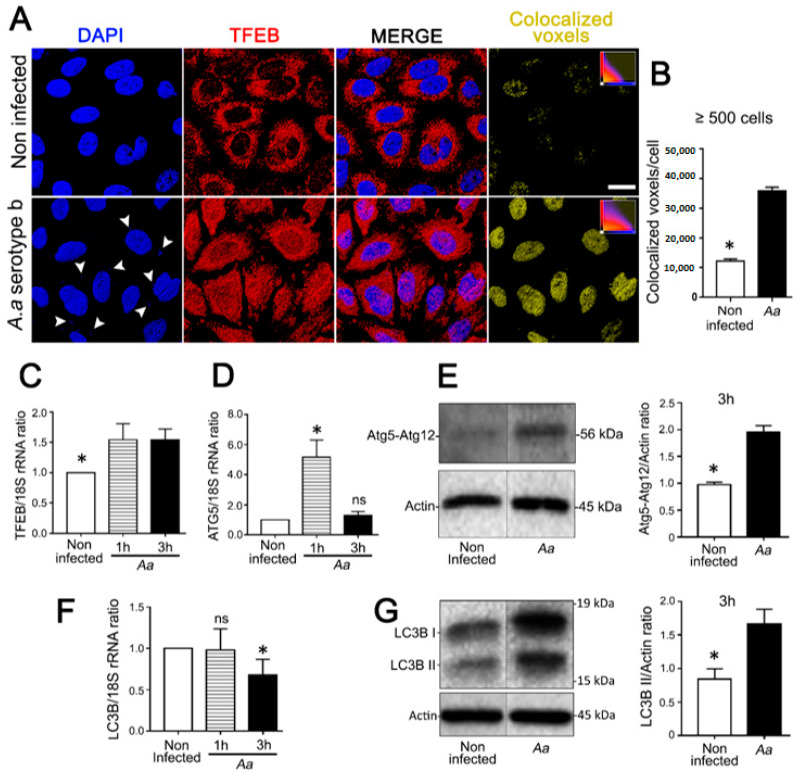
Assessment of autophagy markers in JEKs challenged with *A. actinomycetemcomitans* (*A.a*). OKF6/TERT-2 cells were incubated with *A.a* (serotype b) at MOI = 200 and analyzed as follows: (**A**) Immunofluorescence staining visualized by confocal microscopy. Blue: DNA rich structures stained with DAPI; red: transcription factor EB labeled with anti-TFEB antibody; merge: digital overlap of blue and red channels; yellow: colocalized voxels from merged file. (**B**) Quantification of colocalized voxels is shown on the right (graph). (**C**,**D**,**F**) Relative quantification of transcripts encoding for TFEB (C), ATG5 (D), and LC3B (F) proteins by quantitative real-time PCR normalized to the 18S rRNA transcript levels. (**E**,**G**) Immunoblotting of protein extracts from OKF6/TERT-2 cells incubated with antibodies against the specified autophagy-related proteins. β-actin protein was employed as a loading control. Densitometric quantitation of actin-normalized protein bands is indicated on the right. The results are shown as the average of the actin normalized data from three independent experiments. * *p* < 0.05. Results from (A,E,G) were obtained after 3 h of the bacterial challenge. The vertical dotted lines in (E,G) separate signals that were non-adjacent in the developed membrane.

**Figure 3 cells-09-01221-f003:**
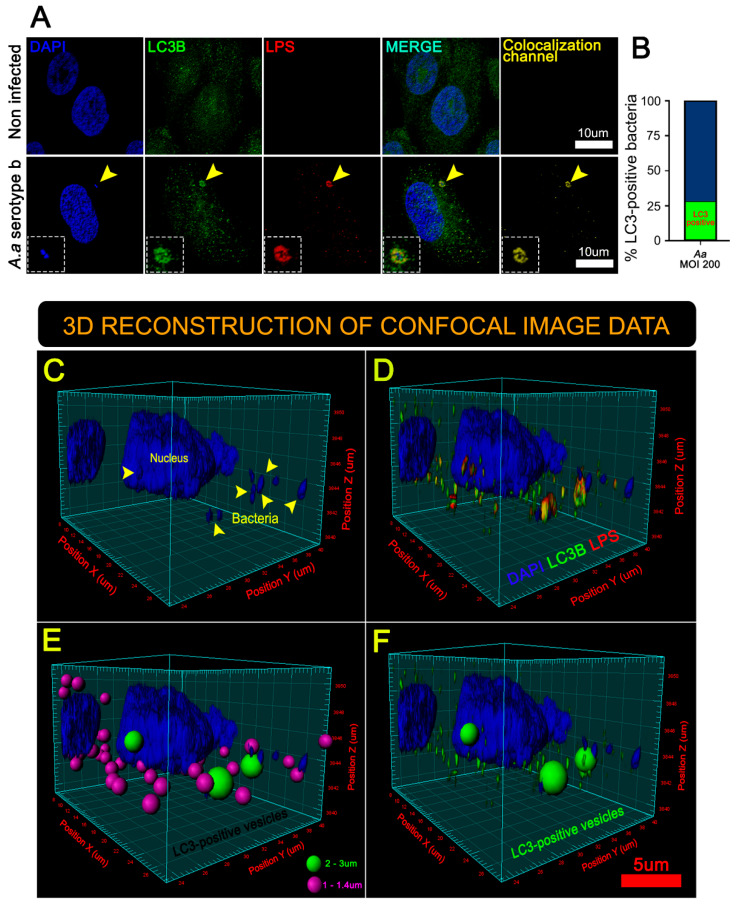
3-D reconstructions of JEKs infected by *A. actinomycetemcomitans* revealed two populations of LC3-positive vesicles. (**A**) Immunofluorescence staining visualized by confocal microscopy. Blue: DNA rich structures stained with DAPI; green: LC3B protein labeled with anti-LC3B antibody; red: *A.a* bacteria labeled with anti-lipopolysaccharides (LPS) antibody; merge: digital overlap of blue, green and red channels; yellow: colocalized voxels from merged file. The yellow arrowhead indicates (area magnified in the insets) a bacteria colocalizing with the host-protein LC3B, scale bar 10 µm. The yellow arrowhead indicates a bacteria colocalizing with the host-protein LC3. (**B**) The graph indicates the fraction of LC3B-positive bacteria (green) related to the total number of intracellular bacteria. A total of ≥1000 analyzed cells (15 microscopic fields) of five different experiments. A series of optical sections (z-stacks) acquired by confocal microscopy were combined to generate a 3-D reconstruction. (**C**) 3-D reconstruction showing the data from the DAPI channel. The yellow arrowhead indicates bacterial DNA. (**D**) 3-D reconstruction showing data from blue (DAPI), green (LC3B), and red (LPS) channels. Note the colocalization of green (LC3B) and red (LPS) channels, seen as yellow hue. (**E**) Same as (D) but emphasizing two kinds of LC3B-positive vesicles; in purple: bacteria-free spheres (1–1.4 µm in diameter), and in green: bacteria-containing spheres (2–3 µm in diameter). (**F**) Same as (E) but highlighting only the large LC3B-positive vesicles containing bacteria (2–3 µm in diameter).

**Figure 4 cells-09-01221-f004:**
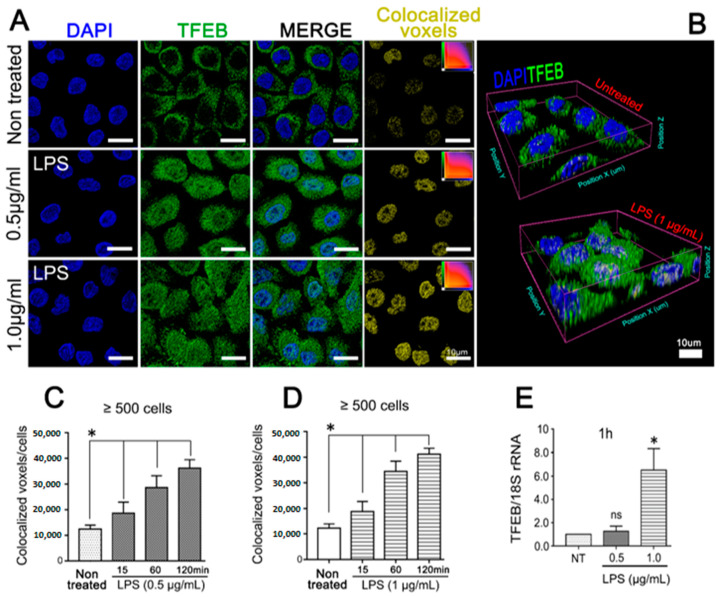
Purified LPS from *A. actinomycetemcomitans* induces TFEB activation. OKF6/TERT-2 cells were incubated with 0.5 or 1 µg/mL of purified LPS from *A.a* (serotype b) and analyzed by (**A**) immunofluorescence visualized by confocal microscopy. Blue: DNA rich structures strained with DAPI; green: transcription factor EB labeled with anti-TFEB antibody; merge: digital overlap of blue and green channels; yellow: colocalized voxels from merged file. (**B**) 3-D reconstruction of confocal image data showing evident TFEB-labeling in LPS-stimulated JEK nuclei, scale bar 10 µm. Quantification of colocalized voxels of LPS-stimulated cells at different time points. (**C**,**D**) Quantification of colocalized voxels from DAPI and TFEB channels in LPS-stimulated cells at different time points. Cells were stimulated with 0.5 (C) or 1 µg/mL (D) of LPS for the indicated period of time. (**E**) Quantitative real-time PCR of TFEB transcripts abundance normalized to 18S rRNA levels after 1 h of LPS stimulation. * *p* < 0.05.

**Figure 5 cells-09-01221-f005:**
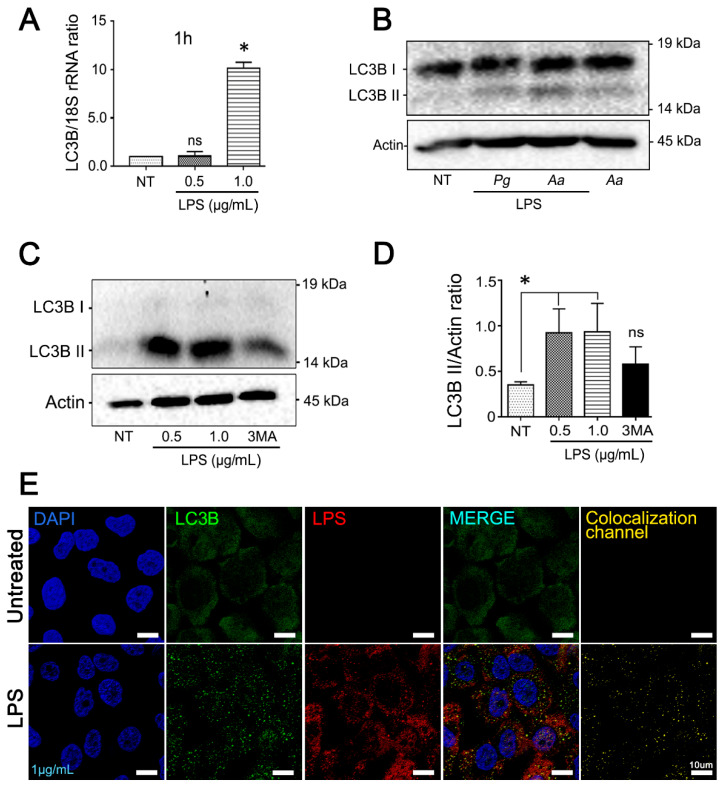
Increased expression of autophagy markers in JEKs stimulated with purified LPS from *A. actinomycetemcomitans*. OKF6/TERT-2 cells were incubated with 0.5 or 1 µg/mL of purified LPS from *A.a*. (**A**) Quantitative real-time PCR of LC3B transcripts abundance normalized to 18S rRNA levels after 1 h of LPS stimulation. (**B**) Immunoblotting of protein extracts from OKF6/TERT-2 cells stimulated for 3 h with LPS or whole *A.a* bacteria incubated with antibodies specific for the autophagy-related protein LC3B. LPS purified from *Porphyromonas gingivalis* (*Pg*) was used as positive control. β-actin protein was employed as a loading control. The results are shown as the average of the actin normalized data from three independent experiments. (**C**) Conversion of LC3B-I to LC3B-II in LPS-stimulated cells. JEKs were pretreated with 3-MA (3 mM) or mock-treated for 3 h before stimulation with LPS. (**D**) Densitometric quantitation of actin-normalized protein bands is indicated on the right. 3-MA used as control. (**E**) Confocal image of JEKs stimulated with 1 µg/mL of LPS. Blue: DNA rich structures stained with DAPI; green: LC3B protein labeled with anti-LC3B antibody; red: LPS purified from *A.a* labeled with anti-LPS antibody; merge: digital overlap of blue, green and red channels; yellow: colocalized voxels from merged file. Note the puncta profile of LC3B-labeling in LPS-stimulated cells. Scale bar 10 µm. * *p* < 0.05.

**Figure 6 cells-09-01221-f006:**
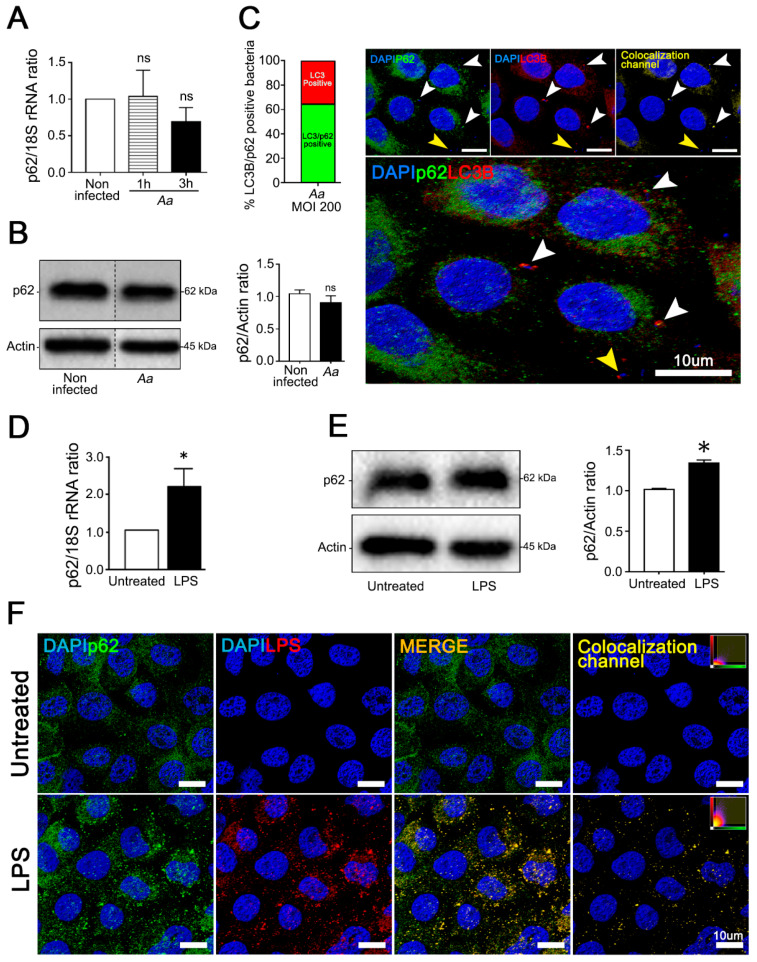
*A. actinomycetemcomitans* (*A.a*) and its LPS induce recruitment of p62 adaptor protein to LC3B-positive vesicles in JEKs. OKF6/TERT-2 cells were incubated with *A.a* or its purified LPS as described. (**A**) Quantitative real-time PCR of p62 transcripts abundance at different time points after bacterial infection normalized to 18S rRNA levels. (**B**) Immunoblotting of protein extracts from infected OKF6/TERT-2 cells incubated with antibodies against the autophagy adaptor protein p62 and β-actin protein as the loading control. Densitometric quantitation of actin-normalized protein bands is indicated on the right. The results are shown as the average of the actin normalized data from three independent experiments. (**C**) Graph: percentage of LC3B/p62-positive bacteria counted in ≥ 500 cells. Image: immunofluorescence of *A.a*-infected cells. Blue: DNA rich structures strained with DAPI; green: autophagy adaptor protein p62 labeled with anti-p62 antibody; red: LC3B protein labeled with anti-LC3B antibody. Colocalization channel: digital overlap of blue, green, and red channels. White arrowheads: LC3B/p62-positive bacteria. Yellow arrowhead: LC3B-positive bacteria. Lower panel: magnified image to enhance visualization of bacterial containing vesicles. (**D**) Quantitative real-time PCR of p62 transcripts abundance after 3 h of LPS stimulation, normalized to 18S rRNA levels. (**E**) Immunoblotting of protein extracts from LPS-stimulated OKF6/TERT-2 cells incubated with antibodies against the autophagy adaptor protein p62 and β-actin protein as a loading control. Densitometric quantitation of actin-normalized protein bands is indicated on the right. (**F**) Immunofluorescence of LPS-stimulated cells visualized by confocal microscopy. Blue: DNA rich structures strained with DAPI; green: autophagy adaptor protein p62 labeled with anti-p62 antibody; red: LC3B protein labeled with anti-LC3B antibody merge: digital overlap of blue, green, and red channels; Colocalization channel: colocalized voxels from merged file. Scale bar 10 µm. ns: non-significant, * *p* < 0.05. The vertical dotted lines in (B) separate signals from non-adjacent lanes in the developed membrane.

**Figure 7 cells-09-01221-f007:**
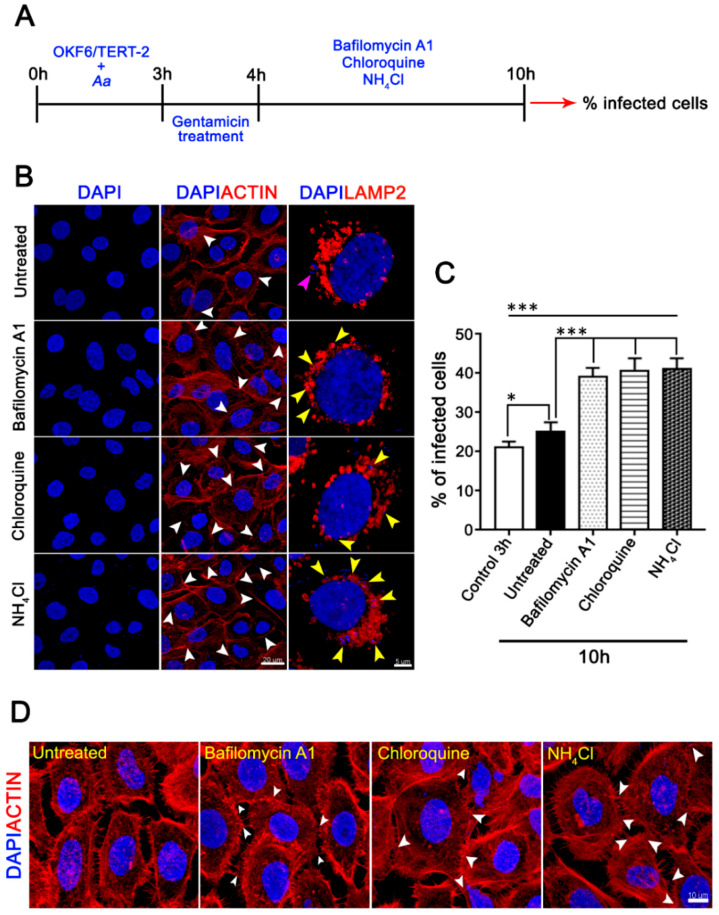
Effect of downstream autophagic inhibitors on host-cell infection by *A. actinomycetemcomitans*. (**A**) Scheme timing showing the experimental design. OKF6/TERT-2 cells were incubated sequentially with *A. actinomycetemcomitans* (MOI = 500), gentamicin to kill non-internalized bacteria, and then with the autophagic inhibitors. (**B**) Fixed and infected cells were labeled with phalloidin-Alexa 568 (red) for visualization of the actin cytoskeleton and DAPI (blue) for visualization of cell nuclei and internalized bacteria (right and central column, white arrowhead indicates infected cell, scale bar 20 µm). Fixed and infected cells immunostained with an anti-LAMP2 antibody for the visualization of lysosomes (orange) and DAPI (left column, scale bar 5 µm). Purple and yellow arrowheads indicate internalized bacteria and internalized bacteria colocalizing closely with the host-cells lysosomes, respectively. (**C**) The graph showed the percentage of infected cells from five independent experiments (≥1000 cells). (**D**) Magnification of confocal images showing bacteria colocalizing with intercellular actin protrusions (white arrowheads, scale bar 10 µm). * *p* < 0.05, *** *p* < 0.001.

**Figure 8 cells-09-01221-f008:**
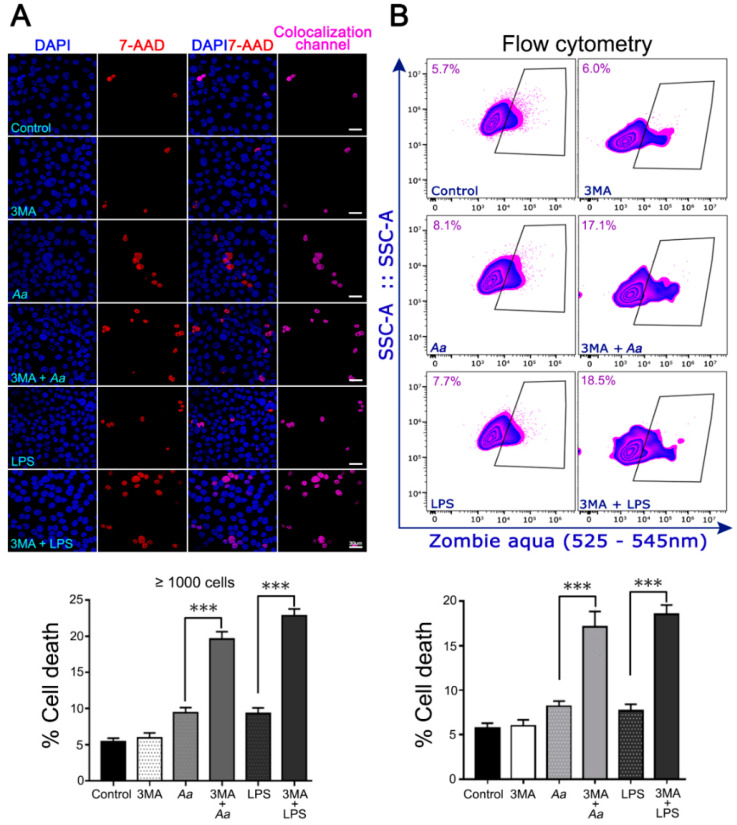
Impact of autophagy pharmacological inhibition on host-cell viability after *A. actinomycetemcomitans* (*A.a*) challenge. OKF6/TERT-2 cells were incubated with *A.a* or its purified LPS in the presence or not of 3-MA, an upstream autophagy inhibitor. (**A**) Fluorescence staining visualized by confocal microscopy. After bacterial infection or LPS-stimulation OKF6/TERT-2 cells were washed and incubated with the fluorescent nucleic acid dye 7-aminoactinomycin (7-AAD). Cells were fixed and stained with the fluorescent dye DAPI and analyzed by confocal microscopy. Blue: DNA-rich structures stained with DAPI (total cell population). Red: DNA-rich structures stained with 7-AAD (dead cells). DAPI/7-AAD: overlap of images obtained from DAPI and 7-AAD channels. Colocalization channel: cells labeled by both fluorophores. The bar graph represents the percentage of dead cells (7-AAD positive) recorded in 1000 cells of three independent experiments. (**B**) *A.a*-infected or LPS-stimulated OKF6-TERT-2 cells were washed and incubated with the amine-reactive fluorescent dye Zombie Aqua and analyzed by flow cytometry. Dot-plots represent the total acquired events showing the live and dead (inside the black box) cell populations. The bar graph below represents the percentage of cell death obtained by flow cytometry of five independent assays. *** *p* < 0.001.

**Figure 9 cells-09-01221-f009:**
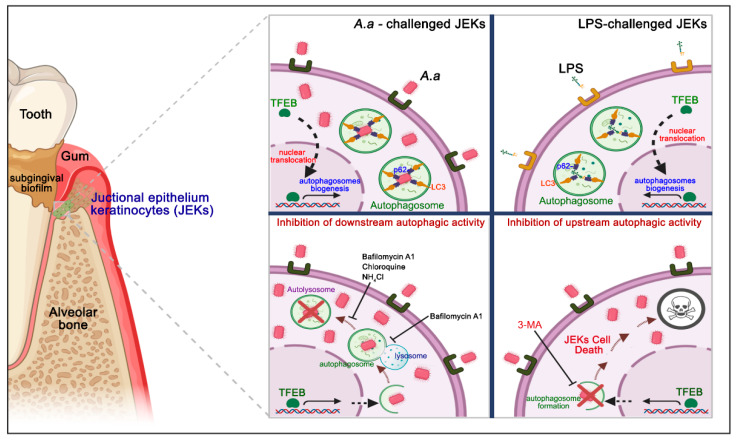
Schematic representation of the junctional epithelium keratinocytes-autophagy induced by *A. actinomycetemcomitans* infection. The junctional epithelium represents the main site of the initial interaction between the dysbiotic subgingival biofilm and host. The challenge and binding of *A. actinomycetemcomitans* and its purified LPS on JEKs surface triggers TFEB translocation to the nucleus and autophagosomes biogenesis. Bacteria and LPS internalized are sequestered to the autophagosomes in formation, through recruitment to the p62-cargo adapter protein, and interact with LC3 protein, favoring autophagy activation (upper panel). Treatment with the alkalinizing compounds of lysosomes bafilomycin A1, chloroquine, and NH_4_Cl, which inhibit the late stages of autophagic flow, induced intracellular bacteria accumulation and significantly increased the infected-JEKs number (lower left panel). Inhibition of autophagosome biogenesis with 3-MA induces cell death in JEKs challenged with *A. actinomycetemcomitans* or its purified LPS in the initial stage of infection (lower right panel). This figure was designed using BioRender.com.

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
