# Peer review of "Aggregatibacter actinomycetemcomitans Induces Autophagy in Human Junctional Epithelium Keratinocytes"

_cells, 2020, doi:10.3390/cells9051221_

Round 1

Reviewer 1 Report

Junctional epithelium keratinocytes (JEKs) serve as an initial line of defense to prevent periodontitis. In the manuscript by Vicencio et al. the authors investigate whether invasion of JEKs by Aggregatibacter actinomycetemcomitans triggers autophagy. They demonstrate that A. actinomycetemcomitans invasion results in TFEB translocation to the nucleus driving an increase in ATG5 RNA, LC3 II protein levels and LC3 puncta. This leads to the authors main conclusion that A. actinomycetemcomitans invasion induces autophagy. The manuscript is in general interesting. However, there are a few concerns about the overall conclusions of the manuscript. One major concern is that the authors conclude that autophagy is upregulated following A. actinomycetemcomitans invasion. However, it is possible that A. actinomycetemcomitans blocks autophagy and this is what leads to the increase in LC3 II protein levels and LC3 puncta. In addition, some experimental details are missing throughout the manuscript.

Concerns –

  1. In Figure 1 and 2 it is very difficult to see the DAPI staining indicating bacterial cells. While the arrows are helpful it is still not convincing that bacterial cells have infected the JEK cells shown. Even when zooming in on the image as much as possible it is very difficult to see the DAPI staining indicating the bacterial cells. The authors should include higher quality images where the bacterial DAPI staining is much clearer or stain for LPS to aid in visualization of the bacteria.
  2. In Figure 2D and 2F two time points are shown post infection. However, in 2E and 2G only one timepoint is shown for Atg5-12 protein levels. In the figure legends no timepoint is stated for the western blots in 2E and 2G. Timepoints should be noted for this figure and related figures throughout the manuscript.
  3. The authors conclude based on the data in Figure 2 that actinomycetemcomitans induces autophagy. However, it is possible that A. actinomycetemcomitans inhibits autophagy instead. Including an assay to monitor autophagy flux would be helpful to indicate whether autophagy is being induced or inhibited in JEK cells following A. actinomycetemcomitans invasion. For example, the authors could use the tandem GFP-RFP LC3 or another similar assay to monitor autophagy flux.
  4. In Figure 3, the authors demonstrate that bacteria colocalize with two different sizes of LC3 rich structures. It would be interesting to see what percent of bacteria colocalize with each size of these puncta. Do bacteria tend to colocalize with smaller or larger LC3 structures more freuqently?
  5. The Y axis is missing in Figure 4 parts C and D. The figure legend for 4C and 4D is also not descriptive enough. It is unclear what the cells were stimulated with.
  6. In Figure 5 it is unclear what the difference is between 5B and 5C. How are these experiments different? Not enough details are provided to understand these differences. How was the assay in 5C performed?
  7. What concentration of LPS was used in Figure 5E?
  8. Figure 6C is very hard to read. The authors should adjust the font of the text in the graph.
  9. In Figure 8 the authors only monitor cell death. Why did the authors also not choose to monitor the % of infected cells as they did in Figure 7?

Author Response

" Please see the attachment "

Reviewer 2 Report

In this study, Vicencio E et al examined the autophagy status in A. actinomycetemcomitans(Aa) infected human junctional epithelium keratinocytes (JEK). Aa or its purified LPS both induced autophagy in JEK. Autophagy of the Aa seems to be mediated via p62 adaptor molecules as in another scenario of xenophagy. Moreover, pharmacological inhibition of autophagy in this Aa xenophagy model increased JEK cell death, suggesting a pro-survival protective role of autophagy. Mechanistically, increased expression of TFEB and its nuclear translocation was noted and hence proposed to induce autophagy in this xenophagy model. 

Overall, the study has been designed and executed reasonably and most of the presented data seem to support the conclusion. However, the mechanistic aspect of how Aa induces autophagy in JEK is weak and needs additional experiments to make a firm conclusion. Endocytosed bacteria in endosomes travel to the lysosome for final degradation. How the bacteria or bacterial components are released into the cytosol to interact with p62 adaptor or LC3 protein, and how these events elevate TFEB expression and nuclear translocation is unclear. Incorporating these mechanistic details will strengthen this study and made it more interesting to the wider readership of the journal.

Furthermore, the following experimental data need to be included to strengthen the study’s conclusion

  • All TFEB related conclusions should be supported by adding TFEB western-blot for total, nuclear or cytosolic level.
  • TFEB is a major transcription factor that upregulates both autophagy-related and lysosomal biogenesis related genes. The authors presented only the expression of LC3 gene. How about the expression level of other autophagy-related and lysosomal biogenesis related genes?
  • All western blots should be continuous for better comparison. Please remove the dotted broken lines in the blot.
  • In Figure 2F: which form of LC3 is elevated? LC3A, 3B, or 3C?
  • In Figure 2C: what does the asterisk mean? The text mentions there are no significant differences in the TFEB level.
  • Figure 2E, 2G, how long the cells were treated with Aa?
  • Figure3A: What is the upper and lower image panels, please label both panels.
  • Figures 6A and 6D: why p62 level is not affected by Aa but LPS treatment increases the p62 expression level?
  • Figure 8: cell death analysis data is presented for the autophagy inhibitor using 3MA. 3MA is an upstream signaling inhibitor of autophagy that could have autophagy off-target effects. Please present similar data for bafilomycin or NH4Cl inhibitor.

Reviewer 3 Report

The authors investigated the role of autophagy in the early junctional epithelium response during the infection with A. actinomycetemcomitans using an in vitro infection model. They used A. actinomycetemcomitans serotype b, its purified LPS, and junctional epithelium keratinocytes to recreate the initial scenario of periodontitis pathogenesis. They concluded that A. actinomycetemcomitans serotype b and its purified LPS induce autophagy that exercises a protective role in the early stage of the infection.

The manuscript is of interest, but in my opinion, it requires some revisions.

Minor revisions:

In the "materials and methods" section

  • The paragraph "in vitro infection model" is poorly detailed. The authors should provide more details about the time-response and dose-response experiments.
  • The use of two different phalloidin-Alexa Fluor-conjugated (488 and 568) and the use of a marker for Tubulin detection should be specified in the "indirect Immunofluorescence confocal assay".

In the "Results" section

  • Fig 1 and Fig 7D should be ameliorated, it is really difficult to visualize bacteria.
  • The authors should rephrase the results showed in figure 2C to make them clearer and in line with the content of the graph.
  • Fig 3A could be ameliorated to enhance the LC3 and LPS immunostaining.
  • Graphs 4C and 4D should be ameliorated, it's difficult to understand what they describe.

I can't open the Supplementary Data file, please could you attach it again?

Round 2

Reviewer 1 Report

The reviewers have addressed all of my concerns. 

Reviewer 2 Report

In this revised version of the manuscript, the authors have partially responded to the “minor “comments. However, the authors still have not addressed one major aspect of the study-How the bacteria or bacterial components are released into the cytosol to interact with p62 adaptor or LC3 protein, and how these events elevate TFEB expression and nuclear translocation?.

This is an important question in terms of the mechanism of how Aa induces autophagy in JEK. The presented data to support the proposed model in Figure 9 is weak. I think it should have been critically discussed in the revised form of the manuscript. The authors have hastily resubmitted the manuscript without considering this important issue.
